# Metabolomics-Based Analysis of Adaptive Mechanism of *Eleutheronema tetradactylum* to Low-Temperature Stress

**DOI:** 10.3390/ani15081174

**Published:** 2025-04-19

**Authors:** Minxuan Jin, Anna Zheng, Evodia Moses Mkulo, Linjuan Wang, Huijuan Zhang, Baogui Tang, Hui Zhou, Bei Wang, Jiansheng Huang, Zhongliang Wang

**Affiliations:** 1College of Fisheries, Guangdong Ocean University, Zhanjiang 524088, China; 2112201066@stu.gdou.edu.cn (M.J.); 2112201120@stu.gdou.edu.cn (A.Z.); evomkulo@gmail.com (E.M.M.); 18813346982@stu.gdou.edu.cn (L.W.); 18303565841@stu.gdou.edu.cn (H.Z.); zjtbg@163.com (B.T.); zhouh89@gdou.edu.cn (H.Z.); wong19820204@126.com (B.W.); fishaqu@126.com (J.H.); 2Guangdong Provincial Key Laboratory of Aquatic Animal Disease Control and Healthy Culture, Zhanjiang 524088, China; 3Agro-Tech Extension Center of Guangdong Province, Guangzhou 510520, China

**Keywords:** metabolomics, cold stress, antioxidant response, liver metabolism, *Eleutheronema tetradactylum*

## Abstract

This study explores the impact of low-temperature stress on *Eleutheronema tetradactylum*, a commercially important fish species. Through comprehensive analyses, we found that cold stress significantly alters liver structure and disrupts metabolic pathways. To cope with these changes, the fish mobilizes its energy metabolism and activates antioxidant systems, demonstrating its ability to adapt to environmental challenges. However, prolonged exposure to cold conditions overwhelms these adaptive mechanisms, leading to liver damage and potential health risks. These findings shed light on the physiological responses of *E. tetradactylum* to cold stress and highlight the importance of understanding such mechanisms in the context of climate change.

## 1. Introduction

Temperature is a critical environmental factor that directly influences the growth, development, and reproduction of fish [1]. Even small fluctuations in water temperature can disrupt cellular homeostasis, compromising physiological functions and overall health. Consequently, water temperature is considered a primary abiotic factor affecting the life processes of fish [2]. The adverse effects of low-temperature stress on fish farming are becoming increasingly significant, particularly due to the challenges posed by climate change and fluctuating water temperatures [3]. As ectothermic organisms, fish rely on their ability to regulate physiological and biochemical processes in response to temperature variations. These adaptive mechanisms include maintaining cellular energy balance, modulating membrane fluidity, and activating antioxidant defenses to mitigate the harmful effects of reactive oxygen species (ROS) [4,5]. Metabolism plays a pivotal role in these adaptive responses by serving as a dynamic interaction between matter and energy, which enables organisms to adjust to environmental stressors [6]. Metabolomics, a powerful tool for the comprehensive and quantitative analysis of metabolites within organisms, allows for the detailed examination of biochemical changes associated with environmental stress, including temperature fluctuations [7]. Recent studies have applied metabolomics to explore cold stress responses in various fish species. For example, low-temperature stress has been shown to significantly affect amino acid metabolism and energy substrates in *Dicentrarchus labrax* [8], while serum metabolomics of black rockfish (*Sebastes schlegelii*) revealed marked alterations in metabolites linked to fatty acid metabolism in response to acute cold stress [9]. These studies not only deepen our understanding of the physiological mechanisms underlying cold stress in fish but also highlight species-specific responses to temperature challenges, in addition to shared adaptive pathways.

*Eleutheronema tetradactylum*, a species native to the Indo-West Pacific, ranges from the Persian Gulf to Papua New Guinea and northern Australia [10]. This species is a warm-temperature, broad-salinity, anadromous migratory fish. Due to its high nutritional value and palatable meat, the scale of its aquaculture has been steadily increasing. However, as a warm-water species, *E. tetradactylum* exhibits limited tolerance to low temperatures. During cold waves, the prolonged decrease in water temperature can severely hinder the growth and survival of *E. tetradactylum*, resulting in significant economic losses for aquaculture operations [11]. Despite the growing importance of *E. tetradactylum* in aquaculture, relatively little research has been conducted on its physiological and biochemical responses to low-temperature stress. Given the increasing frequency of temperature fluctuations and cold spells due to climate change, understanding the mechanisms by which *E. tetradactylum* adapts to these stressors is crucial for improving its resilience and ensuring sustainable aquaculture practices.

The liver is a key metabolic organ in fish, performing essential functions such as detoxification, glycogen storage, and the synthesis of secretory proteins [12]. It plays a central role in the organism’s ability to adapt to environmental stressors [13]. As a hub for metabolic processes, the liver is particularly sensitive to changes in environmental conditions, including temperature fluctuations. In the context of cold stress, the liver’s ability to regulate metabolic pathways is vital for maintaining homeostasis and supporting survival. In this study, non-target liquid chromatography–mass spectrometry (LC-MS) was employed to analyze the metabolite profiles in the liver of *E. tetradactylum* exposed to low-temperature stress. This metabolomics approach offers a comprehensive view of the biochemical changes occurring under stress and provides valuable insights into the mechanisms underlying temperature adaptation. The findings from this study will contribute to a better understanding of the liver’s role in cold stress responses and offer a theoretical basis for strategies aimed at improving the survival rates of fish during winter conditions.

## 2. Materials and Methods

### 2.1. The Experimental Animals

Juvenile *E. tetradactylum* (average body weight: 13.26 ± 2.38 g, average total length: 9.56 ± 0.55 cm) were obtained from a commercial fish farm in Zhanjiang, China. A total of 120 individuals, with similar size, were randomly distributed into six indoor tanks (20 fish per tank). Each tank had a volume of 107 L, with water temperature maintained at 28 °C, dissolve oxygen at 5 mg·L^−1^, pH between 7.6–8.0, and salinity at 28–30.

### 2.2. Experimental Procedures and Samples Collection

The fish were acclimatized to laboratory conditions for one week prior to the temperature experiment. Based on preliminary experiments, the critical low temperature for *E. tetradactylum* was estimated to be 16 °C, and 18 °C was selected for the low-temperature exposure in this study. Fish reared at 18 °C exhibited normal feeding behavior and showed no external signs of abnormal activity. During the acclimation period, the water temperature was gradually reduced by 2 °C per day. After five days, the temperature of the experimental group was stabilized at 18 °C. The temperature-controlled system was maintained using a WN-1C250AN recirculating chiller (Weinuo Experimental Equipment Co. Foshan, China).

Water quality parameters (e.g., ammonia, nitrate, nitrite, and pH) were continuously monitored throughout the experiment, and all tanks were aerated to maintain consistent oxygen saturation levels. A 12:12 light–dark cycle was maintained for the duration of the experiment. The experimental fish were sampled at three time points: 0 days (control), 7 days, and 14 days post-exposure. Three individuals from each tank (18 fish per treatment) were randomly selected for sampling. At each time point, three liver samples were fixed in 4% PFA for histological analysis, while the remaining tissues were snap-frozen and stored at −80 °C. From the frozen batch, three independent samples were randomly selected for antioxidant enzyme assays, and three additional distinct samples were allocated for LC-MS/MS analysis. All residual samples were archived for future studies.

### 2.3. Histological Observation

Fixed tissues were dehydrated through a graded ethanol series (50%, 60%, 70%, 80%, 90%, 95%, 100%). The tissues were then clarified with dimethylbenzene for 30 min and immersed in liquid wax at 60 °C for 60 min. This wax infiltration step was repeated twice. Once infiltrated, the tissues were embedded in paraffin blocks and allowed to cool. Sections of 5 μm thickness were cut using a microtome and mounted onto glass slides. The slides were stained with hematoxylin and eosin (H&E), and digital images were captured using an imaging system for histological analysis. The relative vacuolar area was quantified using ImageJ software (1.54d).

### 2.4. Determination of Antioxidant and Immune Enzyme Activities

Liver tissue samples (0.50 g wet weight) were homogenized in 4.5 mL of 0.86% (*w*/*v*) physiological saline using an IKA^®^ T-18 digital homogenizer (IKA-Werke, Staufen, Germany). The homogenate was centrifuged at 4 °C using a Primo R refrigerated centrifuge (Thermo Fisher Scientific, Dreieich, Germany), and the resulting supernatant was collected for enzymatic analysis. Protein concentration in the enzyme extract was quantified by the Bradford method with Coomassie Brilliant Blue G-250. The activities of total superoxide dismutase (T-SOD), catalase (CAT), glutathione peroxidase (GPx), and malondialdehyde (MDA) content in liver tissues were determined using commercial assay kits (Nanjing Jiancheng Bioengineering Institute, Nanjing, China) following the manufacturer’s instructions.

Statistical analysis was conducted using one-way ANOVA, followed by Duncan’s multiple-range test (SPSS, version 22.0).

### 2.5. Liver Sample Collection and Preparation for LC-MS

Liver tissue samples (*n* = 3 per group) were collected for metabolomic analysis. Immediately after dissection, liver samples were frozen in liquid nitrogen and stored at −80 °C until further processing. Approximately 80 mg of liver tissue was cut on dry ice and transferred into an Eppendorf tube (2 mL capacity). The tissue was homogenized with 200 μL of H_2_O and five ceramic beads using a homogenizer. Subsequently, 800 μL of methanol/acetonitrile (1:1, *v*/*v*) was added to the homogenate for metabolite extraction. The mixture was centrifuged at 14,000× *g* for 15 min at 4 °C. The supernatant was then dried using a vacuum centrifuge, and the residue was re-dissolved in 100 μL of acetonitrile/water (1:1, *v*/*v*) for LC-MS analysis.

### 2.6. LC-MS/MS Analysis

Metabolomic analysis was performed using an ultra-high-performance liquid chromatography system (UHPLC 1290 Infinity LC, Agilent Technologies, Waldbronn, Germany) coupled with a quadrupole time-of-flight mass spectrometer (AB Sciex Triple TOF 6600) at Shanghai Applied Protein Technology Co., Ltd., Shanghai, China. For hydrophilic interaction liquid chromatography (HILIC) separation, a 2.1 mm × 100 mm ACQUIY UPLC BEH 1.7 μm column (Waters, Dublin, Ireland) was used. The mobile phase consisted of 25 mM ammonium acetate and 25 mM ammonium hydroxide in water (A) and acetonitrile (B). The gradient was 85% B for 1 min, reducing linearly to 65% in 11 min, then reducing to 40% in 0.1 min, holding for 4 min, and increasing back to 85% in 0.1 min, followed by a 5 min re-equilibration period.

For reverse-phase liquid chromatography (RPLC) separation, a 2.1 mm × 100 mm ACQUIY UPLC HSS T3 1.8 μm column (Waters, Ireland) was used. In positive ESI mode, the mobile phase contained water with 0.1% formic acid (A) and acetonitrile with 0.1% formic acid (B), and in negative ESI mode, water with 0.5 mM ammonium fluoride (A) and acetonitrile (B). The gradient was 1% B for 1.5 min, increasing linearly to 99% in 11.5 min, followed by 3.5 min at 99% B, and re-equilibrating to 1% B over 0.1 min. A flow rate of 0.3 mL/min was used, with the column maintained at 25 °C. A 2 μL sample was injected per analysis.

The ESI source conditions were set as follows: Ion Source Gas1 (Gas1) at 60, Ion Source Gas2 (Gas2) at 60, curtain gas (CUR) at 30, source temperature at 600 °C, and IonSpray Voltage Floating (ISVF) at ± 5500 V. The mass spectrometer was set to acquire in MS only mode (*m*/*z* range 60–1000 Da) and in auto MS/MS mode (*m*/*z* range 25–1000 Da). The accumulation time for TOF MS scans was set to 0.20 s/spectrum, while product ion scans were acquired with an accumulation time of 0.05 s/spectrum. The collision energy (CE) was fixed at 35 V with ±15 eV, and the declustering potential (DP) was set to 60 V (+) and −60 V (−).

### 2.7. LC–MS Data Processing

Raw MS data were converted into MzXML files using ProteoWizard MSConvert and imported into XCMS software (v3.18.0) for analysis. Peak picking was performed using the following parameters: centWave *m*/*z* = 10 ppm, peak width = c (10, 60), prefilter = c (10, 100). For peak grouping, parameters included bw = 5, mzwid = 0.025, and minfrac = 0.5. CAMERA (Collection of Algorithms of MEtabolite pRofile Annotation) was used for annotation of isotopes and adducts. Only features with more than 50% non-zero measurements in at least one group were retained. Compound identification was based on the accuracy of the *m*/*z* value.

### 2.8. Metabolite Identification and Pathway Analysis

The positive and negative ion data were merged into a combined dataset and imported into the R package ropls (version 1.30.0+) for Orthogonal Partial Least Squares-Discriminant Analysis (OPLS-DA). This model was constructed to examine the relationship between metabolite expression profiles and sample groupings. The OPLS-DA score plot facilitated visualization of sample distribution patterns, while model validity was assessed through permutation testing (n = 2000 iterations) to evaluate potential overfitting. Metabolites were considered significantly altered if their VIP values were >1.0 and *p*-values < 0.05. Pathway enrichment analysis was conducted using the KEGG database (http://www.genome.jp/kegg/pathway.html, accessed during March 2024) to understand the metabolic changes induced by cold stress. Differential metabolites were mapped to metabolic pathways to identify significantly enriched pathways, and a *p*-value < 0.05 was considered statistically significant.

## 3. Results

### 3.1. Histopathological Changes in the Liver Following Hypothermic Stress

To examine the histopathological changes in the liver due to low-temperature stress, we performed histological analysis of liver tissues after 7 and 14 days of exposure to 18 °C. The hepatocytes in both the 7-day and 14-day treatment groups exhibited a polygonal shape, granular cytoplasm, and rounded nuclei, with some histological alterations observed. Notably, the cell boundaries between hepatocytes became less defined and difficult to distinguish after both 7 and 14 days of cold exposure.

Hematoxylin and eosin (H&E) staining revealed that the liver cells in the control group were arranged in a regular pattern with clear cell boundaries and homogeneous cytoplasm (Figure 1a). However, in the 7-day and 14-day experimental groups, fatty degeneration and nuclear atrophy were evident (Figure 1b,c). The vacuolation rate was significantly higher in both the 7-day and 14-day groups compared to the control group (Figure 1d).

### 3.2. Changes in Antioxidant Enzyme Activity in E. tetradactylum Liver

After exposure to low temperature (18 °C) for 7 and 14 days, several antioxidant enzymes in liver tissues were assessed. Compared to the control group, the activities of SOD, CAT, and GPx in the liver significantly increased after 7 days of cold exposure (*p* < 0.05). However, these enzyme activities did not show significant changes after 14 days of exposure (Figure 2a–c). In contrast, the MDA content in the liver significantly increased in both the 7-day and 14-day groups compared to the control group (*p* < 0.05).

### 3.3. Effects of Low Temperature on Metabolomic Alterations in Liver Samples

Liver samples from *E. tetradactylum* were collected after 7 and 14 days of exposure to low temperature (18 °C) and analyzed using untargeted LC–MS metabolomics. As shown in the OPLS-DA score plots (Figure 3a,b and Figure 4a,b), significant separation between the control and experimental groups was observed in both positive and negative ion modes, indicating a substantial effect of low temperature on liver metabolism in *E. tetradactylum*.

In the positive ion mode, the R^2^Y value ranged from 99.7% to 99.8%, and the Q^2^ value ranged from 87.2% to 96.4% (Figure 3a and Figure 4a). In the negative ion mode, the R^2^Y value ranged from 99.8% to 99.9%, and the Q^2^ value ranged from 86.6% to 96.7% (Figure 3b and Figure 4b). The lower values for Q^2^ and R^2^ on the left side of the figures (Figure 3c,d and Figure 4c,d) compared to the original values on the right suggest that the model has excellent fitting and reliable prediction, making it suitable for further analysis of differential metabolites.

### 3.4. Identification of Differential Metabolites

Based on the OPLS-DA results, a total of 87 and 116 distinct metabolites were identified in Con-7d and Con-14d, respectively. In Con-7d, 78 metabolites were upregulated, while 9 metabolites were downregulated. In Con-14d, 97 metabolites were upregulated, and 19 metabolites were downregulated (Figure 5). To visually compare the abundance of metabolites between the cold-treated and control groups, a variable importance in projection (VIP) map was constructed (Figure 6a,b).

### 3.5. Metabolic Pathway Analysis

To explore the metabolic pathways affected by low-temperature stress, pathway enrichment analysis was performed using the KEGG database. The results highlighted several key metabolic pathways that were significantly altered in response to cold stress. For Con-7d, the most relevant metabolic pathways included ABC transporters, carbohydrate digestion and absorption, endocytosis, starch and sucrose metabolism, selenocompound metabolism, and glutathione metabolism (Figure 7a). The most affected pathways of Con-14d were the phosphatidylinositol signaling system, Fc gamma R-mediated phagocytosis, GnRH signaling pathway, fat digestion and absorption, glycerolipid metabolism, and phospholipase D signaling pathway (Figure 7b). The changes in differential metabolites related to differential metabolic pathways are shown in Table 1 and Table 2.

## 4. Discussion

In recent years, the adverse effects of low-temperature stress caused by sudden drops in water temperature have increasingly impacted fish aquaculture [3,14,15]. However, there is a limited understanding of how *E. tetradactylum* responds to cold stress. To address this gap, our study focused on investigating the histological, biochemical, and metabolic responses of *E. tetradactylum* under low-temperature stress. By combining histopathological examination, antioxidant enzyme activity assays, and LC-MS-based metabolomics, we identified key physiological and metabolic alterations that shed light on how this species adapts to suboptimal environmental conditions.

The liver plays a central role in metabolism, excretion, and detoxification, reflecting both the nutritional and pathological status of the organism [16]. Many studies have demonstrated that environmental stressors, including temperature fluctuations, can lead to structural changes in the liver, ultimately impairing its function. Our study found that prolonged cold stress in *E. tetradactylum* resulted in significant liver damage, including tissue vacuolation and nuclear abnormalities. These changes may reflect the excessive energy demands on the liver to maintain basic metabolic functions during cold stress, leading to extensive oxidation and decomposition of hepatic glycogen or lipids. Consequently, the liver’s self-regulatory capacity is compromised, resulting in hepatocyte vacuolation, nuclear atrophy, or disruption, and ultimately causing liver injury. For instance, exposure to nitrite has been shown to cause nuclear hypertrophy and sinusoidal dilation in the liver of bighead carp (*Hypophthalmichthys nobilis*) [17], while exogenous histamine in the diet can induce hepatic inflammation and bleeding in the liver of yellow catfish (*Tachysurus fulvidraco*) [18]. Our study on *E. tetradactylum* demonstrated that prolonged cold exposure (14 days) induced significant oxidative damage, as evidenced by progressively increasing malondialdehyde (MDA) levels—a terminal product of ROS-mediated lipid peroxidation [18]. This persistent MDA accumulation indicates a gradual failure of antioxidant defenses despite their initial activation. Notably, after 14 days of cold stress, we observed a paradoxical decline in hepatic antioxidant enzyme activities (SOD, CAT, GPx) coinciding with peak MDA levels, suggesting complete exhaustion of the compensatory mechanisms under chronic cold stress [19]. This impaired oxidative stress response likely contributes to the observed hepatic damage in *E. tetradactylum*, potentially due to cold-enhanced oxygen solubility disrupting mitochondrial electron transport and exacerbating ROS production [20,21].

Similar response patterns have been documented in other teleosts. In Nile tilapia (*Oreochromis niloticus*), gill tissue MDA concentrations rose significantly within just 24 h at 13 °C [15], while the panfish (*Pampus argenteus*) showed comparable hepatic SOD and CAT activation during early cold exposure [22]. These interspecies parallels suggest conserved vulnerabilities to cold-induced oxidative stress across fish taxa.

At the physiological level, the total antioxidant capacity (T-AOC) integrates enzymatic (SOD/CAT/GPx) and non-enzymatic (GSH, hypotaurine, ascorbate) components to reflect an organism’s oxidative stress resilience [23]. The SOD-CAT system forms the primary defense by converting superoxide radicals to water via hydrogen peroxide detoxification [24]. Our findings reveal that while *E. tetradactylum* initially upregulates these antioxidant pathways (7-day response), the defenses become overwhelmed during prolonged cold exposure, leading to accumulated oxidative damage despite the metabolic costs of maintaining these systems.

Metabolomics analysis provided an integrated view of the metabolic shifts induced by cold stress. A total of 198 differentially expressed metabolites were identified in the experimental groups. Pathway enrichment analysis revealed 12 significantly altered metabolic pathways, highlighting disruptions in energy metabolism, antioxidant defense, and lipid metabolism. At 7 days of cold exposure, the most pronounced changes occurred in pathways related to carbohydrate digestion and absorption, starch and sucrose metabolism, and antioxidant defense, including glutathione metabolism and endocytosis. These pathways are critical for maintaining energy balance and protecting against oxidative stress during cold stress [25,26]. For instance, the increased metabolite levels of maltose in carbohydrate-related pathways suggest that *E. tetradactylum* may rely on glycogenolysis and other carbohydrate metabolism pathways to compensate for energy deficits, similar to other fish species undergoing cold stress [27].

Moreover, the activation of autophagy, a process involved in degrading damaged cellular components and providing energy for cell survival, appears to be crucial for maintaining cellular function under cold stress [28]. This is supported by findings in other fish species, where cold stress induces autophagy to alleviate cellular damage and support energy homeostasis [29]. In line with this, the glutathione antioxidant system also played a key role in mitigating oxidative damage, with significant increases in glutathione levels observed after 7 days of cold stress. This suggests that *E. tetradactylum* employs a robust antioxidant strategy to counteract the harmful effects of ROS under cold conditions, a mechanism also observed in other species like discus fish (*Symphysodon aequifasciatus*), gilthead sea bream (*Sparus aurata*), and pufferfish (*Takifugu fasciatus*) [26,30,31].

The increased levels of glutathione peroxidase (GPx) in the 7-day cold stress group further support the activation of the glutathione metabolic pathway, indicating that *E. tetradactylum* upregulates its antioxidant defense mechanisms in response to oxidative stress induced by low-temperature exposure. In addition, we identified significant alterations in several substrates associated with the ATP-binding cassette (ABC) transporters pathway, such as maltose and glutathione, after 7 days of cold stress. ABC transporters are a large superfamily of proteins involved in diverse physiological processes, including the transport of substrates across cellular membranes in exchange for ATP hydrolysis. These transporters are particularly abundant in tissues such as the liver and kidney, where they mediate critical functions related to metabolism, secretion, and detoxification [32]. The changes observed in the ABC transporter pathway suggest its potential role in modulating the response to cold stress in *E. tetradactylum*, possibly by facilitating the transport of key metabolites involved in stress adaptation.

After 14 days of continuous low-temperature exposure, disruptions in glycerophospholipid metabolism and alterations in the digestion and absorption of fats were observed in the liver. The lipid matrix of biological membranes is composed primarily of glycerophospholipids and sphingolipids, which are essential for maintaining membrane integrity and fluidity [33]. Glycerophospholipids are particularly crucial for maintaining the fluidity of cell membranes and facilitating the transport of energy and substances, enabling them to adapt to environmental changes [34]. In cold conditions, poikilothermic fish must adjust their membrane lipid composition to preserve membrane fluidity and proper cellular function [35]. Our study found a significant increase in phosphatidic acid (PA), a key component of biological membranes, which may reflect a compensatory mechanism to stabilize membrane structures and sustain cellular functions under cold stress.

## 5. Conclusions

This study provides valuable insights into the physiological and metabolic responses of *E. tetradactylum* to low-temperature stress. Through a combination of histological, biochemical, and metabolomic analyses, we demonstrated that cold exposure induces significant changes in liver structure, antioxidant defenses, and metabolic pathways. The mobilization of the carbohydrate and lipid metabolism, particularly through starch and sucrose metabolism, as well as the activation of the glutathione antioxidant system, highlights the fish’s adaptive mechanisms to cope with oxidative stress and energy deficits. The results also suggest that prolonged cold exposure overwhelms the fish’s capacity for oxidative defense and energy metabolism, leading to liver damage. These findings deepen our understanding of how *E. tetradactylum* adapts to cold stress and provide new insights into the physiological mechanisms that could inform aquaculture management practices under fluctuating environmental conditions.

## Figures and Tables

**Figure 1 animals-15-01174-f001:**
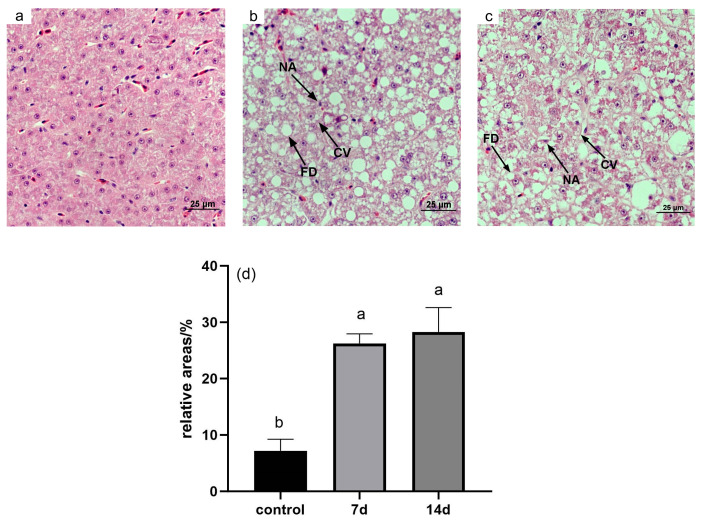
Histological observations of *E. tetradactylum* liver under cold stress. (**a**–**c**) These represent control (28 °C), 18 °C for 7 days, and 18 °C for 14 days, respectively. (**d**) Effect of cold stress on the relative area of hepatic vacuoles in the liver of *E. tetradactylum*. Abbreviations: NA (nuclear atrophy), FD (fatty degeneration), CV (cytoplasmic vacuolization), Letters (a,b) indicate statistically significant differences (*p* < 0.05) determined by ANOVA. Groups sharing no common letter are significantly different.

**Figure 2 animals-15-01174-f002:**
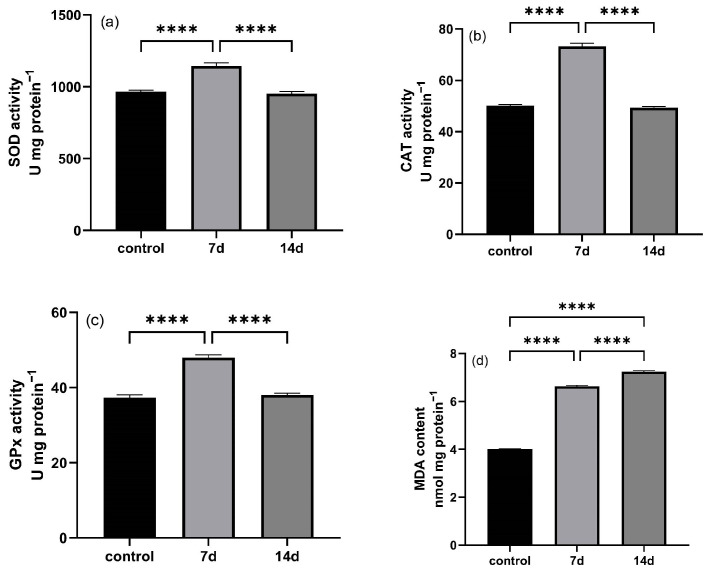
Effect of cold stress on enzyme activities of T-SOD (**a**), CAT (**b**), GPx (**c**), and MDAcontent (**d**) in *E. tetradactylum* liver. Data are expressed as mean ± SEM (n = 3). (**** = *p* < 0.0001).

**Figure 3 animals-15-01174-f003:**
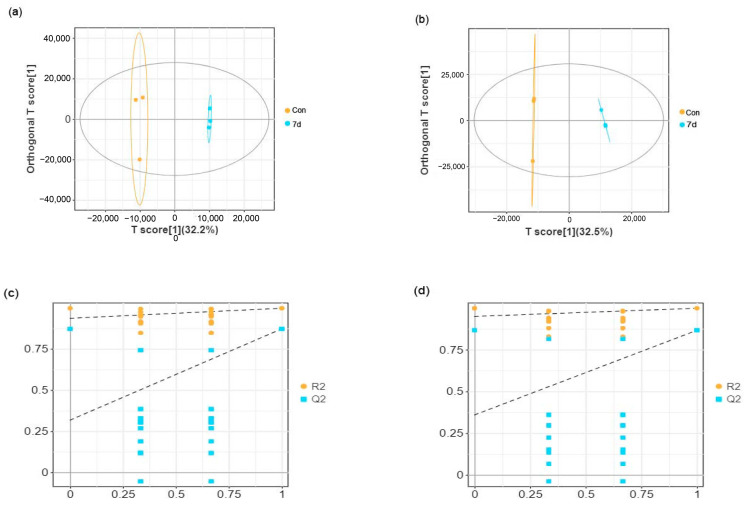
OPLS-DA score plots and model validation for the 7-day cold stress group. (**a**) OPLS-DA score plot in positive ion mode. (**b**) OPLS-DA score plot in negative ion mode. (**c**) Permutation testing of the OPLS-DA model in positive ion mode. (**d**) Permutation testing of the OPLS-DA model in negative ion mode.

**Figure 4 animals-15-01174-f004:**
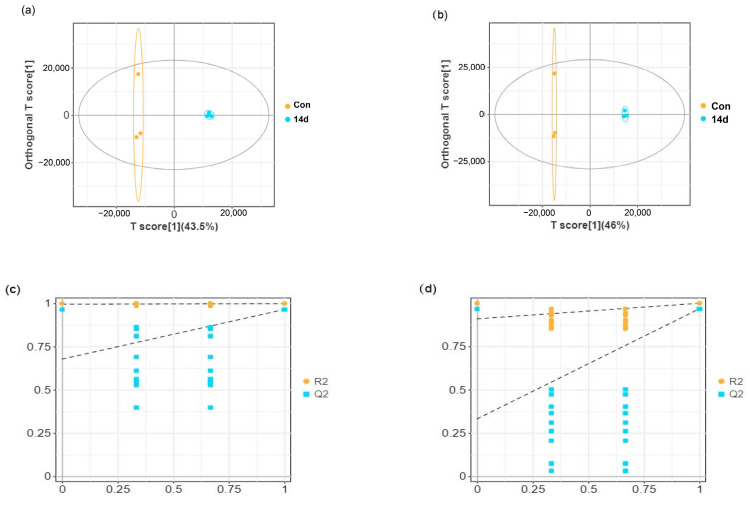
OPLS-DA score plots and model validation for the 14-day cold stress group. (**a**) OPLS-DA score plot in positive ion mode. (**b**) OPLS-DA score plot in negative ion mode. (**c**) Permutation testing of the OPLS-DA model in positive ion mode. (**d**) Permutation testing of the OPLS-DA model in negative ion mode.

**Figure 5 animals-15-01174-f005:**
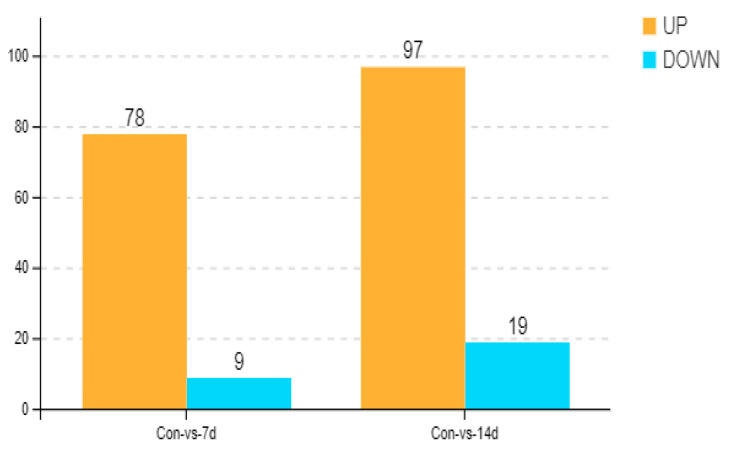
Bar chart showing the upregulation and downregulation of significantly differential metabolites comparing control (Con) vs. 7 days (7 d) and Con vs. 14 days (14 d) of cold stress.

**Figure 6 animals-15-01174-f006:**
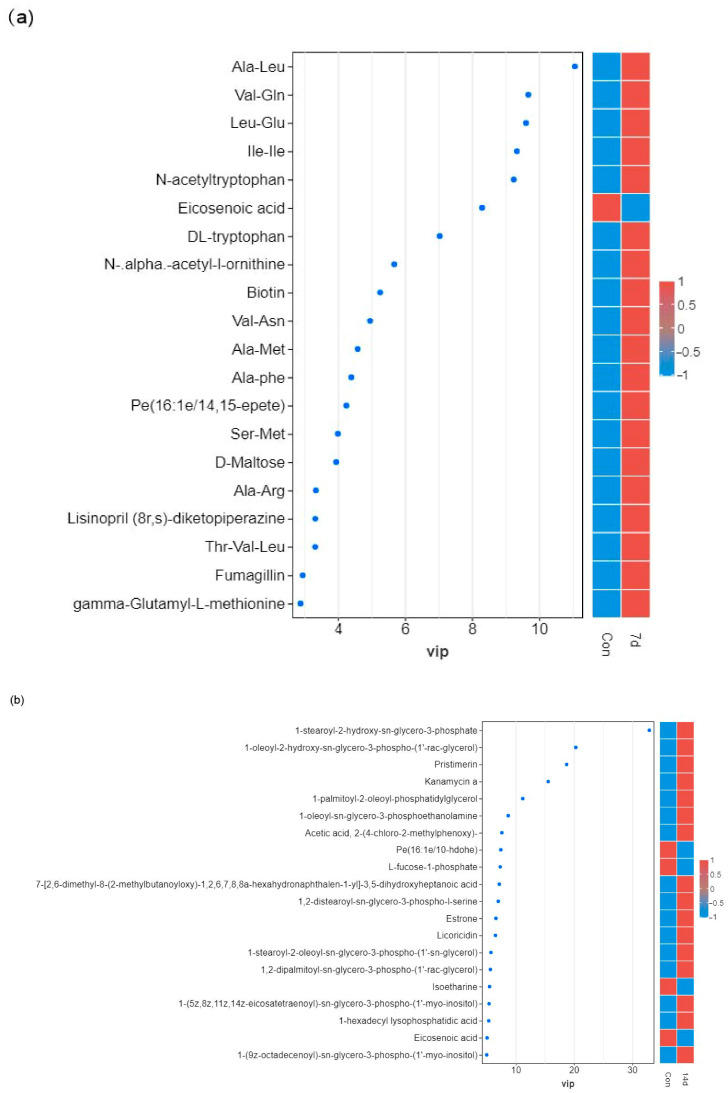
VIP map of top 50 significant discriminating metabolites (SDMs) visualization of differential metabolites in response to cold stress. (**a**) Con vs. 7 days, (**b**) Con vs. 14 days. The color gradient indicates the abundance levels of metabolites, with red representing upregulation and blue representing downregulation.

**Figure 7 animals-15-01174-f007:**
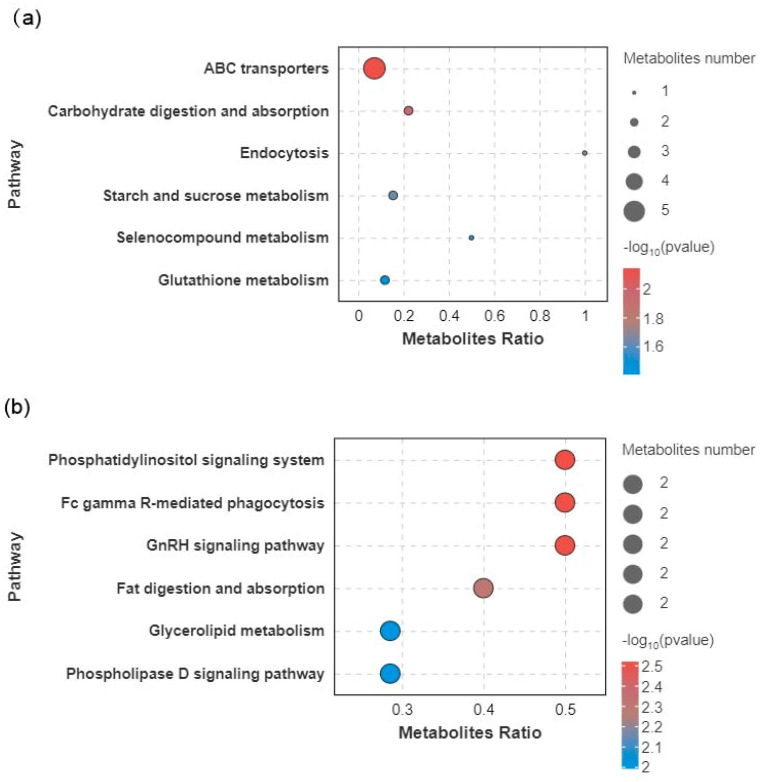
Scatter plot of KEGG pathway enrichment for significant discriminating metabolites (SDMs) between the control and cold stress groups. The size of the dots represents the number of metabolites, and the color indicates the range of the *p*-value. (**a**) Con vs. 7d. (**b**) Con vs. 14d.

**Table 1 animals-15-01174-t001:** The significantly different metabolic pathways and the changes in related metabolites between Con vs. 7d in the liver. Pathways are ordered according to significant level of perturbation (*p* value). Only perturbed pathways with threshold of *p* < 0.05 are shown. SL = significance level (** = very significant with *p* < 0.01, * = moderately significant with *p* < 0.05. Up and down arrows denote upregulated and downregulated metabolites.

Metabolic Pathway	In Set	In Background	*p* Value	SL	Metabolites
ABC transporters	5	70	0.007104	**	Biotin ↑, D-Maltose ↑, Glutathione ↑, L-Valine ↑, Maltose ↑
Carbohydrate digestion and absorption	2	9	0.011049	*	D-Maltose ↑, Maltose ↑
Endocytosis	1	1	0.018764	*	Guanosine 5′-diphosphate ↑
Starch and sucrose metabolism	2	13	0.022904	*	D-Maltose ↑, Maltose ↑
Selenocompound metabolism	1	2	0.037196	*	Seleno-l-methionine ↓
Glutathione metabolism	2	17	0.03821	*	gamma-L-Glutamyl-L-valine ↑, Glutathione ↑

**Table 2 animals-15-01174-t002:** The significantly different metabolic pathways and the changes in related metabolites between Con vs. 14d in the liver. Pathways are ordered according to significant level of perturbation (*p* vales). Only perturbed pathways with threshold of *p* < 0.05 are shown. SL = significance level (** = very significant with *p* < 0.01, * = moderately significant with *p* < 0.05. Up and down arrows denote upregulated and downregulated metabolites.

Metabolic Pathway	In Set	In Background	*p* Value	SL	Metabolites
Phosphatidylinositol signaling system	2	4	0.002988	**	1-Hexadecanoyl-2-(9Z-octadecenoyl)-sn-glycero-3-phosphoric acid ↓, 1-stearoyl-2-linoleoyl-sn-glycero-3-phosphate ↑
Fc gamma R-mediated phagocytosis	2	4	0.002988	**	1-Hexadecanoyl-2-(9Z-octadecenoyl)-sn-glycero-3-phosphoric acid ↓, 1-stearoyl-2-linoleoyl-sn-glycero-3-phosphate ↑
GnRH signaling pathway	2	4	0.002988	**	1-Hexadecanoyl-2-(9Z-octadecenoyl)-sn-glycero-3-phosphoric acid ↓, 1-stearoyl-2-linoleoyl-sn-glycero-3-phosphate ↑
Fat digestion and absorption	2	5	0.00491	**	1-Hexadecanoyl-2-(9Z-octadecenoyl)-sn-glycero-3-phosphoric acid ↓, 1-stearoyl-2-linoleoyl-sn-glycero-3-phosphate ↑
Glycerolipid metabolism	2	7	0.010026	*	1-Hexadecanoyl-2-(9Z-octadecenoyl)-sn-glycero-3-phosphoric acid ↓, 1-stearoyl-2-linoleoyl-sn-glycero-3-phosphate ↑
Phospholipase D signaling pathway	2	7	0.010026	*	1-Hexadecanoyl-2-(9Z-octadecenoyl)-sn-glycero-3-phosphoric acid ↓, 1-stearoyl-2-linoleoyl-sn-glycero-3-phosphate ↑

## Data Availability

The original contributions presented in this study are included in the article. Further inquiries can be directed to the corresponding author(s).

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
