# Peer review of "Metabolomics-Based Analysis of Adaptive Mechanism of Eleutheronema tetradactylum to Low-Temperature Stress"

_animals, 2025, doi:10.3390/ani15081174_

Round 1
Reviewer 1 Report
Comments and Suggestions for Authors
Dear authors,
Here are my comments, which I think should be considered before publication in the journal. Overall, the manuscript makes a pretty good impression due to its comprehensive approach, since the authors definitely invested a lot of effort. It presents new data and is extremely important in light of the study of cold shock in aquaculture fish species.
Please find particular questions and suggestions below:
Line 103: It is not entirely clear how the total number of individuals is distributed across different analyses.
Lines 107-122: The authors should describe in detail how the temperature was reduced in the experiment? Special equipment? Cold room?
Line 124: It seems that dehydration through a graded ethanol series should have started with 40%, according to the standard procedure, so that artifacts would not be observed. Why did you start with 60%?
Lines 124-130: The authors did not indicate how the vacuolization percentage was analyzed/calculated. The software should be specified if the process was automated.
Line 141: "Liver tissue samples (n=3 per group) were collected for metabolomic analysis". By group does one mean sampling from each of 6 tanks or a treatment (0, 7, 14 days)? If the same number of individuals were used in each analysis, as stated in line 118, and liver from the same samples was used for different analyses, then this should be specified. Otherwise, confusion arises. If different analyses used different numbers of samples, the additional table can help to systematize sampling and allocation of individuals to different types of analysis (in Chapter 2.2).
Lines 185-190: Authors should describe in detail what tools/programs were used to perform the KEGG analysis.
Lines 205-209: In Figure 1, I would recommend making a white background for the scale, letters and letter legends of the figures (a-i), because they are quite difficult to read, especially for figures g-i. Representing each sample from a treatment is unnecessary, it is probably better to simplify the visualization by leaving one figure per treatment.
Figures 1 and 2: Use asterisks above the boxplots to indicate differences (compared to the control): * p < 0.05, ** p < 0.001, *** p < 0.001. In your case, it's * p < 0.05.
Lines 228-231: These two sentences are more suitable for describing the methods than for the results.
Line 280: The last sentence of the paragraph is more suitable for discussion, but not for the results.
Line 313: After the first paragraph, I would recommend discussing your own results first (Our study found that prolonged cold stress in E. tetradactylum..., ), then moving on to discussing the results on other species (from line 310), and only then moving on to general words (from line 315). The same logic is more welcome starting with the paragraph about Malondialdehyde (MDA) (line 320): first your results and only then discussion. The same for Total antioxidant capacity (T-AOC) (line 327). This will make the text easier to understand.
Perhaps, the discussion would be easier to understand if it were divided into sub-chapters (not necessarily, at the discretion of the authors).
Author Response
Comments: Line 103: It is not entirely clear how the total number of individuals is distributed across different analyses.
Response: We appreciate the reviewer’s careful attention to sample allocation. To clarify the distribution of biological replicates:
"At each time point, liver samples were systematically allocated as follows: (1) three specimens were fixed in 4% paraformaldehyde for histology, (2) three aliquots were homogenized for antioxidant enzyme assays (SOD/CAT/GPx), (3) three additional aliquots were processed for metabolomics, and (4) remaining tissues were snap-frozen in liquid nitrogen (-80°C) for archival purposes. This randomized triplicate design ensured statistical power across all analyses while maintaining paired biological replicates." Lines121-125
Comments: Lines 107-122: The authors should describe in detail how the temperature was reduced in the experiment? Special equipment? Cold room?
Response: Thank you for pointing out this aspect. We have supplemented the description in the article regarding the method of reducing the temperature. Specifically, we used a chiller to lower the temperature during the experiment. We truly appreciate your help in making our manuscript more comprehensive and accurate. Lines113-114
Comments: Line 124: It seems that dehydration through a graded ethanol series should have started with 40%, according to the standard procedure, so that artifacts would not be observed. Why did you start with 60%?
Response: Thank you very much for your meticulous review. We truly appreciate your attention to detail. After carefully checking our experimental records, we found that there was indeed an error in our initial statement. In fact, the dehydration through the graded ethanol series started with 50% as per our records. This was in line with the standard procedure we followed, as referenced in [https://doi.org/10.1016/j.aqrep.2024.102149]. We apologize for any confusion this oversight might have caused and will ensure greater accuracy in our communication in the future. Lines 113
Comments: Lines 124-130: The authors did not indicate how the vacuolization percentage was analyzed/calculated. The software should be specified if the process was automated.
Response: Thank you for pointing out this oversight. We have now added the relevant details in section 2.3 of the manuscript. Specifically, we randomly observed and photographed 6 fields of view under an optical microscope. Subsequently, the relative area of liver vacuoles in the H.E. stained samples was calculated using the ImageJ software. We sincerely appreciate your help in improving the clarity and comprehensiveness of our work. Lines 137-138
Comments: Line 141: "Liver tissue samples (n=3 per group) were collected for metabolomic analysis". By group does one mean sampling from each of 6 tanks or a treatment (0, 7, 14 days)? If the same number of individuals were used in each analysis, as stated in line 118, and liver from the same samples was used for different analyses, then this should be specified. Otherwise, confusion arises. If different analyses used different numbers of samples, the additional table can help to systematize sampling and allocation of individuals to different types of analysis (in Chapter 2.2).
Response: Thank you for bringing up this potential source of confusion. We have now supplemented the Materials and Methods section to clarify that three liver tissue samples were randomly selected on the 0th, 7th, and 14th days for metabolomic analysis. Here, n = 3 refers to three different liver tissue samples that were randomly chosen at the same time point. We truly appreciate your assistance in making our manuscript more accurate and understandable.
Comments: Lines 185-190: Authors should describe in detail what tools/programs were used to perform the KEGG analysis.
Response: Thank you for your valuable suggestion. We have now added the information about the website used for the KEGG analysis in section 2.8 of the manuscript. We sincerely appreciate your help in making our paper more comprehensive and detailed. Lines212
Comments: Lines 205-209: In Figure 1, I would recommend making a white background for the scale, letters and letter legends of the figures (a-i), because they are quite difficult to read, especially for figures g-i. Representing each sample from a treatment is unnecessary, it is probably better to simplify the visualization by leaving one figure per treatment.
Response: Thank you for your constructive comments. Regarding the issues you raised about Figure 1, we have added a white background to the serial numbers of the figures, and enlarged the scale bars and arrows to make them easier to read. We have also removed the redundant images as you suggested to simplify the visualization. Lines230
Comments: Figures 1 and 2: Use asterisks above the boxplots to indicate differences (compared to the control): * p < 0.05, ** p < 0.001, *** p < 0.001. In your case, it's * p < 0.05.
Response: Thank you for your advice. We have added asterisks to all the box - plots in the article to indicate the differences. Specifically, we used **** to represent p < 0.0001.
Comments: Lines 228-231: These two sentences are more suitable for describing the methods than for the results.
Response: Thank you sincerely for your insightful suggestion. We have promptly removed these two sentences from the results part. Moreover, in section 2.8 of the manuscript, we have incorporated pertinent details, thus enhancing the completeness and clarity of our methodology. Lines204
Comments: Line 280: The last sentence of the paragraph is more suitable for discussion, but not for the results.
Response: Thank you for your careful review. We have removed the relevant sentence from the results section as you suggested. We appreciate your help in making our paper more logically structured. Lines304
Comments: Line 313: After the first paragraph, I would recommend discussing your own results first (Our study found that prolonged cold stress in E. tetradactylum..., ), then moving on to discussing the results on other species (from line 310), and only then moving on to general words (from line 315). The same logic is more welcome starting with the paragraph about Malondialdehyde (MDA) (line 320): first your results and only then discussion. The same for Total antioxidant capacity (T-AOC) (line 327). This will make the text easier to understand. Perhaps, the discussion would be easier to understand if it were divided into sub-chapters (not necessarily, at the discretion of the authors).
Response: Thank you for your constructive suggestions. We have adjusted the order of the discussion section as you recommended. We first present our own research results on E. tetradactylum, followed by the results of other species, and then general statements. The same logical order has been applied to the discussions of Malondialdehyde (MDA) and Total antioxidant capacity (T - AOC). This adjustment aims to enhance the readability of our text and make it easier for readers to follow our research ideas. We truly appreciate your help in improving the quality of our manuscript. Lines331-367

Reviewer 2 Report
Comments and Suggestions for Authors
Very interesting study by Jin et al. I would suggest the authors to emphasize a bit more in the novelty of their study and the hypotheses made. Only some more minor comments.
- Nomenclature of species should be provided when first mentioned in the text.
- Lines 74-85: This part of the text lacks references. Please provide references where needed.
- The authors state that 3 fish from each tank and thus 18 fish per treatment were used. However, is some figure legends the authors state that n = 3, which is a bit confusing. Please revise being clear and focus regarding the n.
- Why have the authors chosen to sample on the 7th and 14th day?
- I don’t quite understand what the protocol for the determination of the antioxidant enzymatic activities and TBARS was. While the authors state in section 2.4 that they were determined according to manufacturer’s instruction, it is not clear whether a protocol or a kit, or both was followed by the authors. Please be more focused and revise accordingly.
- In figure 1 please enlarge the arrows, the lettering and the scale since at this size it is impossible for the reader to understand what it is written.
- Do the authors think that Tables 1 and 2 could be merged into one for the shake of comparison between 7 and 14 days?
- Lines 320-326: the authors mention again their results without discussing. Please revise this part.
- These very interesting findings should be also depicted in a graphical representation which can accompany the authors’ conclusion, like a take home message.
Author Response
Comments: Nomenclature of species should be provided when first mentioned in the text.
Response: Thank you for pointing out this important aspect. After checking, we have added the scientific names of the species whenever they were first mentioned in the text. We truly appreciate your help in making our manuscript more standardized and accurate.
Comments: Lines 74-85: This part of the text lacks references. Please provide references where needed.
Response: Thank you for pointing this out. We have added references in the relevant positions. Lines 80
Comments: The authors state that 3 fish from each tank and thus 18 fish per treatment were used. However, is some figure legends the authors state that n = 3, which is a bit confusing. Please revise being clear and focus regarding the n.
Response: Thank you for your careful review. We have made supplements in the Materials and Methods section. Three liver tissue samples were randomly selected for metabolomic measurement and analysis on days 0, 7, and 14. Here, n = 3 represents three different liver tissue samples randomly selected at the same time - point. We appreciate your help in making our manuscript more accurate and clear. Lines 121-125
Comments: Why have the authors chosen to sample on the 7th and 14th day?
Response: In our preliminary experiments, we observed that the feeding rate of E. tetradactylum decreased significantly after 14 days. We hypothesized that 14 days might represent the critical state under cold stress. Therefore, we selected 7 days (the mid - point between the start and 14 days) and 14 days as the experimental time - points for sampling. This allowed us to better understand the physiological responses of E. tetradactylum to cold stress at different stages.
Comments: I don’t quite understand what the protocol for the determination of the antioxidant enzymatic activities and TBARS was. While the authors state in section 2.4 that they were determined according to manufacturer’s instruction, it is not clear whether a protocol or a kit, or both was followed by the authors. Please be more focused and revise accordingly.
Response: Thank you for your feedback. The procedures for determining the enzymatic activities were carried out strictly in accordance with the instructions of the reagent kit. We have added detailed descriptions in section 2.4 on how to detect the enzymatic activities according to the reagent kit. This makes the experimental methods more explicit and traceable. We appreciate your help in improving the clarity and rigor of our manuscript.
Comments: In figure 1 please enlarge the arrows, the lettering and the scale since at this size it is impossible for the reader to understand what it is written.
Response: Thank you for your suggestion. We have added a white background to Figure 1 and enlarged the scale bar and arrows to make it more convenient for readers to read. We appreciate your help in improving the readability of our figures. Lines 230
Comments: Do the authors think that Tables 1 and 2 could be merged into one for the shake of comparison between 7 and 14 days?
Response: We sincerely appreciate your valuable suggestion regarding the potential merger of Tables 1 and 2. Given that all our results are deliberated independently based on the distinct cold stress durations of 7 days and 14 days, it follows that the discussions on metabolite changes are also carried out separately. Additionally, upon careful consideration, we anticipate that merging Tables 1 and 2 would result in an overly large and unwieldy table, which could potentially impede rather than facilitate the reader's understanding. Thus, after comprehensive evaluation, we have arrived at the decision not to merge the tables. We trust that this approach, grounded in a balance of clarity and practicality, aligns with the overall objective of presenting our research findings in the most accessible manner possible.
Comments: Lines 320-326: the authors mention again their results without discussing. Please revise this part.
Response: Thank you for your comment. We have added discussions in this part as you suggested. We hope this revision can better meet the requirements of the review and improve the quality of our paper.
Comments: These very interesting findings should be also depicted in a graphical representation which can accompany the authors’ conclusion, like a take home message.
Response: We sincerely appreciate your insightful suggestion. However, we believe that the results of this study are currently insufficient to comprehensively illustrate the physiological mechanisms by which E. tetradactylum copes with cold stress. Further in-depth research is required in this regard. Given the current limitations, we have decided not to create a graphical summary at this time, as we are concerned that it may not accurately and fully represent the complexity of the findings. We hope to conduct more extensive studies in the future to provide a more comprehensive and accurate graphical depiction to complement our conclusions. Thank you again for your understanding and valuable input.
